# Is impaired energy production a novel insight into the pathogenesis of pyridoxine-dependent epilepsy due to biallelic variants in *ALDH7A1*?

**Anastasia Minenkova[1], Erwin E. W. Jansen[2], Jessie Cameron[3,4], Rob Barto[2], Thomas Hurd[1], Lauren MacNeil[3,5], Gajja S. Salomons[2,6], Saadet Mercimek-Andrews[5,7,8] \***

1 Department of Molecular Genetics, University of Toronto, Toronto, Ontario, Canada, 2 Metabolic Unit, Department of Clinical Chemistry, Amsterdam UMC, Vrije Universiteit Amsterdam, Amsterdam Neuroscience, Amsterdam Gastroenterology & Metabolism, Amsterdam, The Netherlands, 3 Metabolic Laboratory, Department of Pediatric Laboratory Medicine, The Hospital for Sick Children, Toronto, Ontario, Canada, 4 Department of Pediatric Laboratory Medicine, University of Toronto, Toronto, Ontario, Canada, 5 Department of Medical Genetics, University of Alberta, Edmonton, Alberta, Canada, 6 Laboratory Genetic Metabolic Diseases, Department of Clinical Chemistry, Amsterdam UMC, University of Amsterdam, Amsterdam Neuroscience, Amsterdam Gastroenterology & Metabolism, Amsterdam, The Netherlands, 7 Division of Clinical and Metabolic Genetics, Department of Pediatrics, University of Toronto, Toronto, Ontario, Canada, 8 The Hospital for Sick Children, Toronto, Ontario, Canada

\* saadet@ualberta.ca

**Data Availability Statement:** All relevant data are within the manuscript and its Supporting information files.

## Abstract

### Background

Pyridoxine-dependent epilepsy (PDE) is due to biallelic variants in *ALDH7A1* (PDE-*ALDH7A1*). *ALDH7A1* encodes α-aminoadipic semialdehyde dehydrogenase in lysine catabolism. We investigated the gamma aminobutyric acid (GABA) metabolism and energy production pathways in human PDE-*ALDH7A1* and its knock-out *aldh7a1* zebrafish model.

### Methods

We measured GABA pathway, and tricarboxylic acid cycle metabolites and electron transport chain activities in patients with PDE-*ALDH7A1* and in knock-out *aldh7a1* zebrafish.

### Results

We report results of three patients with PDE-*ALDH7A1*: low paired complex I+II and complex II+III and individual complex IV activities in muscle biopsy in patient 1 (likely more severe phenotype); significantly elevated CSF glutamate in the GABA pathway and elevated CSF citrate, succinate, isocitrate and α-ketoglutarate in the TCA cycle in patient 3 (likely more severe phenotype); and normal CSF GABA pathway and TCA cycle metabolites on long-term pyridoxine therapy in patient 2 (likely milder phenotype). All GABA pathway metabolites (γ-hydroxybutyrate, glutamine, glutamate, total GABA, succinic semialdehyde) and TCA cycle metabolites (citrate, malate, fumarate, isocitrate, lactate) were significantly

**Funding:** SMA was funded by the Department of Pediatrics at the University of Toronto. The funders had no role in study design, data collection and analysis, decision to publish, or preparation of the manuscript.

**Competing interests:** The authors have declared that no competing interests exist.

low in the homozygous knock-out *aldh7a1* zebrafish compared to the wildtype zebrafish. Homozygous knock-out *aldh7a1* zebrafish had decreased electron transport chain enzyme activities compared to wildtype zebrafish.

## Discussion

We report impaired electron transport chain function, accumulation of glutamate in the central nervous system and TCA cycle dysfunction in human PDE-*ALDH7A1* and abnormal GABA pathway, TCA cycle and electron transport chain in knock-out *aldh7a1* zebrafish. Central nervous system glutamate toxicity and impaired energy production may play important roles in the disease neuropathogenesis and severity in human PDE-*ALDH7A1*.

## Introduction

Pyridoxine-dependent epilepsy (PDE) (OMIM#266100) is due to biallelic variants in *ALDH7A1* (PDE-*ALDH7A1*) [1–3]. α-Aminoadipic semialdehyde (α-AASA) dehydrogenase (EC 1.2.1.31) in the lysine catabolic pathway is encoded by *ALDH7A1* [NM_001201377.2 (variant 1); GI: 1676439670; NM_001202404.2 (variant 2) GI: 1677703364] and its deficiency causes accumulation of α-AASA, $\Delta^1$- piperideine-6-carboxylate (P6C), and pipecolic acid (PA). P6C inactivates pyridoxal-5'-phosphate [3, 4]. Recently a new biomarker, called 6-oxopiperidine 2 carboxylic acid (6-oxo-pipecolate), was identified using a stable isotope-labeled internal standard, and nonderivatized liquid chromatography tandem mass spectrometry-based method. The authors report that α-AASA undergoes intramolecular cyclization without losing water and converts to P6C/piperidine-6-hydroxy-2-carboxylate (6-hydroxy-pipecolate) [5].

About 200 patients with PDE-*ALDH7A1* were reported in the literature [6–11]. Its estimated incidence was approximately 1:64,352 live births in a recent study [8]. All patients with biallelic pathogenic or likely pathogenic variants have elevated α-AASA/P6C in body fluids. PA was elevated in most of the patients with biallelic pathogenic or likely pathogenic variants in *ALDH7A1* [9]. The recent consensus guidelines recommend that the biomarkers α-AASA and P6C can be used alone or in combination with other biomarkers to increase sensitivity and specificity [9]. Seizures, developmental delay, and cognitive dysfunction are the common clinical features of PDE-*ALDH7A1*. Seizure onset is usually in the neonatal or early infantile period. Seizure freedom is reported in about 75% of patients on pyridoxine therapy [12]. Normal neurocognitive functions with borderline impairments in various processing functions have been reported in 25% of the patients on pyridoxine therapy [12].

Recently, we reported a well characterized knock-out *aldh7a1* zebrafish model [13]. Our knock-out *aldh7a1* zebrafish has a homozygous 5 base pair (bp) deletion in *aldh7a1*. Homozygous knock-out *aldh7a1* zebrafish embryos have spontaneous rapid circling swim behavior after 9 days post fertilization (dpf) and die before 14 dpf. Homozygous knock-out *aldh7a1* zebrafish embryos have elevated α-AASA, P6C and PA [13].

Seizures and neurodevelopmental delays are likely multifactorial in PDE-*ALDH7A1*: 1) Accumulation of α-AASA and P6C due to the block at α-aminoadipic semialdehyde dehydrogenase is presumed to be neurotoxic; 2) Pyridoxal-5'-phosphate is the active pyridoxine metabolite, which is used as a cofactor for about 70 biochemical reactions in the human body [14]. Its deficiency has detrimental effects to the human body presenting with several biochemical abnormalities and enzyme dysfunctions; 3) Gamma aminobutyric acid (GABA) is the

principal inhibitory neurotransmitter in the brain. Enzymes in the GABA metabolism require pyridoxal-5-phosphate as a cofactor. It is likely that GABA production is decreased, which causes seizures due to the perturbed balance of neuronal excitation; 4) Lysine catabolism provides acetyl-coenzyme A (acetyl-CoA) and GABA metabolism provides succinate to tricarboxylic acid (TCA) cycle. Decreased substrate supply to the TCA cycle due to PDE-*ALDH7A1* may affect energy production pathways. All these factors could contribute to ongoing seizures and/or neurodevelopmental delays in PDE-*ALDH7A1*.

To investigate the effects of PDE-*ALDH7A1* on the vitamin B6 metabolism, GABA metabolism and energy production, we measured cerebrospinal fluid (CSF), GABA pathway and TCA cycle metabolites in two patients with PDE-*ALDH7A1* using their remaining stored CSF samples, which were collected during their clinical management. We report electron transport enzyme activity measurements in a patient with PDE-*ALDH7A1*. Additionally, we measured vitamin B6 vitamers, GABA pathway and TCA cycle metabolites, and electron transport chain enzyme activities in the knock-out *aldh7a1* zebrafish embryos at the time of spontaneous rapid circling swim behavior and compared to their age-matched wildtype zebrafish embryos.

## Materials and methods

### Patients

This study was approved by The Hospital for Sick Children's Research Ethics Board (REB) approval number 1000050808. The corresponding author (S.M-A) followed all patients with PDE-*ALDH7A1* in her neurometabolic clinic at The Hospital for Sick Children and was in the circle of clinical care. There were 12 patients followed in this clinic between January 2012 and November 2020 by the corresponding author. All patients with PDE-*ALDH7A1* were diagnosed and/or referred between 2000 and 2018 to this clinic for management. Eleven of those patients with PDE-*ALDH7A1* were diagnosed between 2000–2017 and reported for their phenotypes, genotypes, and treatment outcomes as part of a retrospective cohort study in 2017 (REB#1000050808). We reviewed their electronic patient charts (consent was exempted for the retrospective chart review study) [12]. Only one patient with PDE-*ALDH7A1* was diagnosed between 2017–2020 and reported recently as part of a large patient cohort of childhood epilepsy [15]. Out of 12 patients, there was only one patient who had electron transport chain enzyme activity measurement in muscle and was included into this study (patient 1). The parent signed a case report consent form to allow us to include the child's information into the current study. There were only two patients (patient 2 and patient 3) who had remaining CSF samples and were included into this study. Both patients were reported previously [12, 16, 17]. GABA pathway and TCA cycle metabolites were measured in the remaining stored CSF samples of two patients, which were collected during their clinical management. Those CSF samples were stored in the Clinical Metabolic Laboratory, Department of Clinical Chemistry, Amsterdam after the completion of their CSF α-AASA and PA metabolite measurements as part of their treatment outcome monitoring. An informed consent was signed by parents and/or patients to allow us to use the remaining CSF samples for the measurement of GABA pathway and TCA cycle metabolites and report the results.

Additionally, we contacted the Canadian Inherited Metabolic Diseases Research Network (CIMDRN), which is a multidisciplinary research network including investigators from 14 different metabolic genetic centers from across Canada. Between 2015 and 2020, there were 11 patients with PDE-*ALDH7A1* enrolled into this research study. Six of those 11 patients were enrolled by the corresponding author (S.M-A) from her neurometabolic clinic described above. The remaining 5 patients were enrolled from four different metabolic genetic centers in Canada. Corresponding author (S. M-A) contacted physicians who enrolled their patients

with PDE-*ALDH7A1* to CIMDRN from those four centers. All four physicians reported that none of those five patients underwent muscle electron transport chain enzyme activity measurements nor had remaining CSF sample for us to include into our current research study. Physicians from those four centers reported that they have been following additional three patients with PDE-*ALDH7A1* in their center, but again, none of those additional three patients had muscle electron transport chain enzyme activity measurements nor a remaining CSF sample for us to include into our current research study. In total, there were 20 patients from five different metabolic genetic centers in Canada, and these were the only three patients with PDE-*ALDH7A1* from our center that we were able to enroll into our study, as they either had muscle electron transport chain enzyme activity measurements or a remaining CSF sample for the measurement of GABA pathway and TCA cycle metabolites.

## Zebrafish

Zebrafish (wild-type and heterozygous *aldh7a1)* were housed, maintained, and bred at The Hospital for Sick Children Zebrafish Core Facility. The study was approved by The Hospital for Sick Children Animal Care Committee, (ACC#41617). Details of model development using CRISPR-Cas9 mutagenesis, genotype, behavior analysis, electrophysiology studies, survival and biochemical features of knock-out *aldh7a1* zebrafish model was previously reported by the corresponding author's research group (S.M-A) [13]. Experiments were performed on embryos (at or <12 dpf). Zebrafish were monitored daily over 10–15 minutes for any seizure onset from the 9 dpf and then returned to the main system. All embryos with seizure onset and same age wildtype were anesthetized using Tricaine and sacrificed on ice for metabolite measurements with minimalized pain and suffering. All zebrafish procedures were performed in compliance with the Animals for Research Act of Ontario and the Guidelines of the Canadian Council on Animal Care and were carried out in strict accordance with the approved conditions to ameliorate animal suffering.

Heterozygous 5 bp knock-out *aldh7a1* zebrafish females and males were crossed, and produced eggs were held in a petri dish until they started swimming as an embryo; they were then placed into the system water for feeding at 5 days post fertilization (dpf). They were fed according to the general feeding schedule as per the Zebrafish Core Facility rules. Due to the large number of embryos required for various metabolite measurements, we always used 3–4 small tanks and generated all embryos at once. The embryos generated from those zebrafish were grown until they showed seizure-like locomotor behavior. Embryos with seizure-like locomotor behavior were genotyped using previously reported high resolution melt analysis to confirm homozygosity prior to metabolite measurement [13]. After genotypic confirmation, 20 homozygous 5 bp knock-out *aldh7a1* zebrafish embryos were collected into one tube for metabolite measurements. Same strain wildtype zebrafish females and males were crossed at the same time and were grown until the same age as the collected homozygous knock-out *aldh7a1* zebrafish. Sacrificed embryos were stored at -80˚C until metabolite measurement of all metabolites.

## Vitamin B6 vitamers

Pyridoxal-5'-phosphate, pyridoxamine-5'-phosphate, pyridoxal, pyridoxine, and pyridoxamine were purchased from Sigma-Aldrich. Four deuterated vitamers were used as internal standards. $d_2$-Pyridoxal 5-phospate•$H_2O$ was kindly supplied as a gift by S.P. Coburn, Department of Chemistry, Indiana University-Purdue University (Fort Wayne, IN). $d_2$-Pyridoxine was purchased from Buchem BV (Apeldoorn, The Netherlands). $d_3$-Pyridoxal•HCl was synthesized by Beta Chem (Leawood, KS) and $d_2$-Pyridoxine•HCl was purchased from CDN

Isotopes (CDN Isotopes, Canada). Heptafluorobutyric acid was purchased from Sigma. Acetic acid was bought from Merck and trichloroacetic acid was purchased from VWR Chemicals.

Zebrafish homogenates for homozygous and wildtype groups were prepared using 20 embryos each. Vitamin B6 vitamers were measured using a previously reported method for CSF [18] and wildtype served as controls. Samples were diluted with water two and ten times to achieve linear range of the calibration curves. Diluted samples (100 μL) were deproteinized by mixing with 1 mL 5% (w/v) trichloroacetic acid containing isotopically labeled internal standards and measured using a series LC-30AD UPLC system (Shimadzu) coupled to an API 5000 triple-quadrupole tandem mass spectrometer (Applied Biosystems Sciex). The mass spectrometer was operated in the positive-ionization mode.

We injected 20 μL of supernatant into a stable-bond C8 reversed phase column (150 $^{x}$ 4.6mm, 3.5μm; Agilent). The mobile phase consisted of (A) 3% (v/v) acetic acid with 4 mM heptafluorobutyric acid and (B) 90% acetonitrile with 4 mM heptafluorobutyric acid. A flow rate of 0.2 mL/min was maintained and the vitamin B6 vitamers were eluted using the following program 0–0.2 min isocratic hold 0% B; 0.2–4.5 min linear gradient 0–25% B; 4.5–8 min linear gradient 25–80% B; 8–9 min linear gradient 80–0% B and 9–12 min isocratic 0% B. Pyridoxamine-5'-phosphate was quantitated using $d_2$-Pyridoxal 5-phospate as internal standard and isotopically labeled $d_2$-Pyridoxine was also used for pyridoxamine. The method was linear for pyridoxal-5'-phosphate and pyridoxal from 0.4–100 nM, and from 0.1–40 nM for pyridoxamine-5'-phosphate, pyridoxine and pyridoxamine. Analyst 1.6.3 software (ABSciex) was used for data acquisition and analysis.

## GABA pathway and TCA cycle metabolites

Zebrafish homogenates for homozygous and wildtype groups using 20 embryos were kept on ice and sonicated four times for 10 seconds in 250 μL of distilled water. The lysates were then centrifuged at 13000 rpm for 10 min at 4ºC. Supernatant was removed for metabolite measurements. L-[$^{13}$C5] glutamic acid (Isotec/Sigma-Aldrich, St Louis, MO, USA), stable-isotope-labelled $^{13}$C4 succinic semialdehyde (prepared in house using previously reported method by Struys et al., 2005), $d_2$-GABA (MSD Isotopes (Montreal, Canada) and $^{2}$H$_6$-gamma hydroxybutyrate, $^{2}$H$_4$-citrate (Euriso-Top), $^{13}$C$_4$-fumarate (Cambridge Isotope Laboratories), $^{13}$C$_4$-succinate (Cambridge Isotope Laboratories), $^{2}$H$_3$-L-malate (Milipore Sigma), $^{2}$H$_4$- α-ketoglutarate (Euriso-Top), and $^{13}$C-L-lactate (Milipore Sigma) were used as internal standards.

GABA pathway metabolites were measured according to previously reported methods [19]. GABA was measured using electrospray ionization negative ion mode [11]. Succinic semialdehyde was measured by liquid chromatography-tandem mass spectrometry (LC-MS/MS) according to Struys et al. [20].

TCA cycle metabolites, citrate, isocitrate, L-malate, fumarate, succinate, α-ketoglutarate (α-KG) and lactate levels, were measured by LC-MS/MS according to Blom et al. [21] with some modifications. Briefly, to 20 μL of zebrafish sample a mixture of 0.2 nmol $^{2}$H$_4$-citrate, 0.2 nmol $^{13}$C$_4$-fumarate, 0.2 nmol $^{13}$C$_4$-succinate, 0.4 nmol $^{2}$H$_3$-L-malate, 0.2 nmol $^{2}$H$_4$- α-ketoglutarate and 2 nmol of $^{13}$C-L-lactate was added as internal standard. To each sample 20 μL H2O was added. The mixtures were filtered by Milipore Ultrafree centrifugal filters, Durapore- pVDF 0.22.

Samples were mixed and 6 μL of the sample was injected onto the LC-MS/MS. For the quantification of citrate and isocitrate, the following multiple reaction monitoring transitions were used: citrate m/z -191/-87, $^{2}$H$_4$-citrate m/z -195/-89, isocitrate m/z -191/-73, succinate m/z -117/-73, $^{13}$C$_4$-succinate m/z -121/-76, fumarate m/z -115/-71, $^{13}$C$_4$-fumarate m/z -119/-74, L-malate m/z -133/-115, $^{2}$H$_3$-L-malate m/z -136/-117, α-ketoglutarate m/z -145/-57,

$^2H_4$- α-ketoglutarate m/z -149/-60, lactate m/z -89-43 and $^{13}$C-L-lactate m/z -90/-44. Sciex API 5000 Instrument settings for TCA cycle metabolites were summarized in S1 Table in S1 File. The optimization of LC-MS/MS system was summarized in S2 Table in S1 File. Age matched controls for CSF metabolite measurements were banked after the completion of their clinical investigations in the Metabolic Unit, Department of Clinical Chemistry, Free University, Amsterdam. Those samples were de-identified by clinical biochemist (please see acknowledgements) to serve as control. The control data was fully anonymized before shared with research team and study principal investigator. Co-authors E.E.W. Jansen and G.S. Salomons have been involved in several studies applying the measurement of GABA pathway and TCA cycle metabolites [20, 22–25] and have extensive knowledge for the measurement of these metabolites.

### Electron transport chain enzyme activity measurements

Pooled batches of 300 homozygous knock-out *aldh7a1* zebrafish and age matched wildtype zebrafish embryos were prepared for mitochondrial isolation. Pools were homogenized, mitochondria were isolated by differential centrifugation, and pellets were stored at -80˚ until testing [26]. Activity measurements of isolated complex I, complex II, complex IV and complex V and paired complex I+III and complex II+III electron transport chain complexes were performed spectrophotometrically using previously described protocols [26–28]. Citrate synthase activity was determined spectrophotometrically as described [29]. Enzyme activities were normalized to protein content determined by the Lowry method and expressed as nmol/min/mg mitochondria. All these measurements were performed by one technician providing clinical and research measurements longer than five years.

### Mitochondrial DNA qPCR

Two independent crosses were performed to generate 50 *aldh7a1* knock-out and 50 wildtype zebrafish embryos. A sample size of 50 embryos for mtDNA extraction has been previously shown to adequately represent mtDNA amount of the population [30]. Embryos (10dpf) with seizure-like locomotor behavior and age matched wildtype embryos were batched in groups of five in microfuge tubes. DNA extraction was performed by isopropanol precipitation using a previously reported method [31]. DNA concentrations were determined using NanoDrop 1000 Spectrophotometer (Thermo Scientific).

qPCR was performed using 8ng of total DNA and 300nM of primers. Primers with efficiency >98% for mtDNA and with >94% for nDNA were used for subsequent analysis. To account for pipetting errors, Ct values from mtDNA primers were normalized to the Ct values from nDNA primers (primers are listed in S1 Table in S1 File). Each sample was run in triplicate. qPCR was performed with FroggaBio SensiFast no ROX qPCR master mix (BIO-98050) using BioRad C1000 Touch CFX384 real-time thermal cycler. The PCR program was: 2 min at 95˚C, 45 cycles of 95˚C for 5 sec and 60˚C for 30 sec. Melt curves generated through a thermal denaturation step were used to verify primer amplification specificity. The resulting ΔCt values were the differences of the averages of Ct values between mtDNA primer sets and nDNA primer sets. The resulting $2^{-\Delta Ct}$ values were normalized to mean mtDNA amount from wildtype embryos. The primers used in this study are listed in S3 Table in S1 File.

### Statistical analysis

Data for vitamin B6 vitamers, GABA metabolism pathway and TCA cycle metabolites were analyzed using unpaired multiple t-tests followed by Holm-Sadak correction for multiple analysis (with α = 0.05) to determine statistical significance. Data for mitochondrial DNA amount

were analyzed using unpaired 2-tailed student t-test. All statistical analysis was performed in Graphpad Prism software (version 7.0). Statistical analysis was performed between average wildtype and homozygous knock-out values.

## Results

### Patients

**Patient 1.** This 12.5-year-old girl was diagnosed with PDE-*ALDH7A1* at the age of 7 months. She was born at 33 weeks of gestation by C-section. She was admitted to the neonatal intensive care unit (NICU) for 3 weeks. She had her first afebrile generalized tonic-clonic seizure at the age of 3.5 months. Her EEG showed intermittent spike waves and sharp waves over the right temporal region. She was treated with phenobarbital and phenytoin. She continued having frequent generalized tonic-clonic seizures and clobazam and levetiracetam were added to her seizure treatment. She presented with status epilepticus at the age of 5 and 7 months and her EEG showed multifocal frequent spikes. Due to refractory epilepsy, she underwent a surgical open muscle biopsy and a lumbar puncture at the age of 6 months. Muscle histopathology and electron microscopy were normal. Electron transport chain activities were low for paired (complex I+II and complex II+III) and individual (complex IV) activities, which are summarized in Fig 1 and S4 Table in S1 File. Her CSF monoamine neurotransmitter analysis was suggestive of PDE-*ALDH7A1* which led to pyridoxine treatment at the age of 7 months. Her diagnosis was confirmed by direct Sanger sequencing of *ALDH7A1* (homozygous IVS-12 (+1) G>A in *ALDH7A1*) at the age of 11 months. She has been seizure free on pyridoxine (200 mg/day) therapy since the diagnosis. She was started on arginine supplementation (12 g/day) and protein-restricted diet and was initially compliant to both therapies but became non-complaint later. She has moderate intellectual disability (patient 9 in reference [12]). We did not

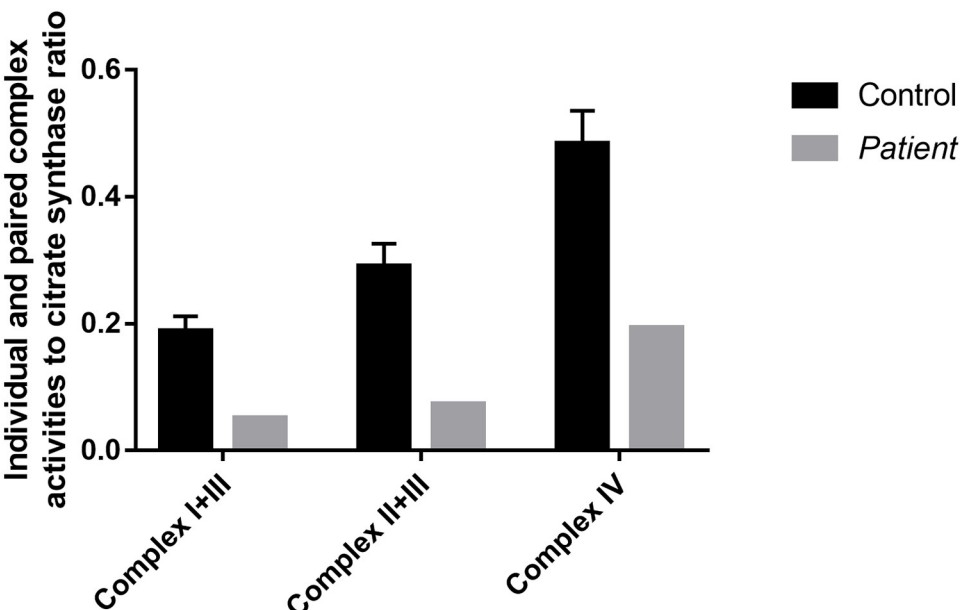

**Fig 1. Electron transport chain paired and individual complexes.** Electron transport chain paired and individual complexes in muscle biopsy specimen are depicted, which shows decreased paired and individual complex activities in Patient 1 compared to controls.

have a remaining CSF sample to measure GABA pathway and TCA cycle metabolites in patient 1.

**Patient 2.** This 19-year-old young man had focal seizures at the age of 2.5 months. Pyridoxine was started at the age of 3 months. He had a clinical diagnosis of PDE due to seizure freedom on pyridoxine monotherapy. He had elevated urine α-AASA (8.8 mmol/mol creatinine; reference range 0–0.5) and compound heterozygous variants [(c.500A>G (p.Asn167Ser) and c.1481+1G>T] in *ALDH7A1* confirming the diagnosis of PDE-*ALDH7A1* at the age of 11 years [17]. He has been seizure free on pyridoxine (200 mg/day) therapy and has been on arginine supplementation (12 g/day) with good compliance (patient 4 in reference [9]). He has normal intellectual functions [5]. We had his residual CSF sample (collected at 11 years of age) from his previous lumbar puncture on pyridoxine therapy (200 mg/day) which was stored at -80˚C. We summarized his CSF GABA pathway and TCA cycle metabolite results in Tables 1 and 2. All GABA pathway metabolites were normal compared to age matched controls. Fumarate and α-ketoglutarate were not detectable in Patient 2 and age matched controls.

**Patient 3.** This 8.5-year-old boy presented with generalized clonic seizures at age 4 days. Due to refractory epilepsy, he was treated with phenobarbital, phenytoin, lorazepam, midazolam. He was started on pyridoxine at age 21 days. He had a confirmed diagnosis of PDE-*ALDH7A1* by elevated urine α-AASA (39.6mmol/mol creatinine; reference range 0.0–2) (at the age of 2.5 months) and compound heterozygous variants (c.446C>A (p.Ala149Glu) and c.919C>T (p.Arg307X)) in *ALDH7A1* (at the age of 4 months) [16]. He has been seizure free on pyridoxine (200 mg/day) therapy and has been on arginine supplementation (12 g/day) and protein-restricted diet (good compliance) (patient 8 in reference [9]). He has normal intellectual functions, but gross motor delay [9]. We had his residual CSF sample from his previous lumbar puncture at age 2 weeks prior to pyridoxine therapy and at age 7 months on pyridoxine therapy (200 mg/day) monotherapy which were stored at -80˚C. We summarized his CSF GABA pathway and TCA cycle metabolite results in Tables 1 and 2. Glutamate was significantly elevated in the 2-week-old CSF sample and moderately elevated in the 7-month-old

**Table 1. GABA pathway metabolites of Patient 2 and Patient 3 with PDE-*ALDH7A1*.**

| GABA metabolites | Patient 2 µmol/L (on pyridoxine therapy) | Age matched controls for Patient 2 µmol/L | Patient 3 (prior to pyridoxine therapy) µmol/L | Patient 3 µmol/L (on 6-months of pyridoxine therapy) | Age matched controls for Patient 3 µmol/L* |
|---|---|---|---|---|---|
| Glutamine | 437 | 352–680[a] | 468 | 523 | 388–824[f] |
| Glutamate | 2.0 | 1–13[b] | *170* | *97.0* | 1–11[g] |
| Total GABA | 6.8 | 3.2–10.5[c] | 8.1 | 13.5 | 4.5–14.2[h] |
| γ-hydroxybutyrate | 0.68 | 0–2.6[d] | 0.26 | 0.31 | 0–2.6[e] |
| Succinic semialdehyde | 0.02 | 0.01–0.04[e] | 0.04 | 0.01 | <0.01–0.02[i] |

Italic and bold indicated outside of the reference range.

[a] >3 years

[b] >3 years

[c] >10 years

[d] 0–16 years

[e] 6–16 years

[f] 0–1 years

[g] 0–1 years

[h] 0–10 years

[i] 0–1 years

  

**Table 2.  TCA cycle metabolites of Patient 2 and Patient 3 with PDE-*ALDH7A1*.**

| TCA cycle metabolites | Patient 2 μmol/L (on pyridoxine therapy) | Age matched controls for Patient 2 μmol/L (6–16 years) | Patient 3 μmol/L (prior to pyridoxine therapy) | Patient 3 μmol/L (on 6-months of pyridoxine therapy) | Age matched controls for Patient 3 μmol/L (0–1 year) |
|---|---|---|---|---|---|
| **Citrate** | 239 | 190–288 | 231 | ***269*** | 188–249 |
| **Succinate** | 2.65 | 2.06–2.65 | 2.54 | ***1.64*** | 2.5–3.34 |
| **Malate** | 1.21 | 0.94–2.18 | ***4.41*** | 2.07 | 1.4–3.17 |
| **Fumarate** | ND | ND | ND | ND | ND |
| **Isocitrate** | 11.8 | 11.3–14.1 | ***11.4*** | ***19.8*** | 8.7–10.3 |
| **Lactate** | 1156 | 854–1314 | 1042 | 1054 | 886–1394 |
| **α-ketoglutarate** | 0 | ND | ***35.5*** | ***5.16*** | 1.25 |

**Abbreviations:** wks = week(s); mo = month(s); yr = year(s); TCA = tricarboxylic acid; ND = not detectable

Italicized and bold indicates metabolite levels were outside of the reference range.

CSF sample compared to age matched controls. All other GABA pathway metabolites were normal compared to age matched controls. Fumarate was not detectable in patient 3 and in age matched controls. Malate, isocitrate and α-ketoglutarate were elevated in the 2-week-old CSF sample compared to age matched controls. Citrate, succinate, isocitrate and α-ketoglutarate were elevated in the 7-month-old CSF sample compared to age matched controls.

## Zebrafish

All embryos from incrossed heterozygous 5 bp deletion females and males were placed in system water for feeding from 5 dpf according to our institutional zebrafish feeding guidelines. Between 10 to 11 dpf, all embryos with a rapid circling swim behavior, followed by loss of posture were homozygous for 5 bp deletion.

## Vitamin B6 vitamers

To investigate vitamin B6 vitamers, we measured pyridoxal-5'-phosphate, pyridoxamine-5'-phosphate, pyridoxal, pyridoxine, and pyridoxamine levels. Pyridoxal-5'-phosphate, pyridoxamine-5'-phosphate, pyridoxal and pyridoxamine levels were significantly lower in the knock-out *aldh7a1* zebrafish embryos compared to the wildtype (Fig 2). All metabolite measurements are depicted in S5 Table in S1 File. Pyridoxine level trended lower in the knock-out *aldh7a1* embryos (p = 0.0979) but was not statistically significant.

## GABA pathway metabolites

To investigate GABA metabolism, we measured the levels of glutamine, glutamate, GABA, γ-hydroxy-butyrate, and succinic semialdehyde in knock-out *aldh7a1* and wildtype zebrafish. The levels of all metabolites were significantly lower in knock-out *aldh7a1* zebrafish embryos compared to the wildtype (Fig 3). All metabolite measurement results are depicted in S5A Table in S1 File.

## TCA cycle metabolites

To investigate whether TCA cycle could be impaired in knock-out *aldh7a1* zebrafish, we measured citrate, isocitrate, succinate, fumarate, malate, and lactate. All metabolite measurement results are depicted in S5B Table in S1 File. There were significantly lower levels of all metabolites in knock-out *aldh7a1* zebrafish relative to the control wildtype zebrafish (Fig 4A & 4B).

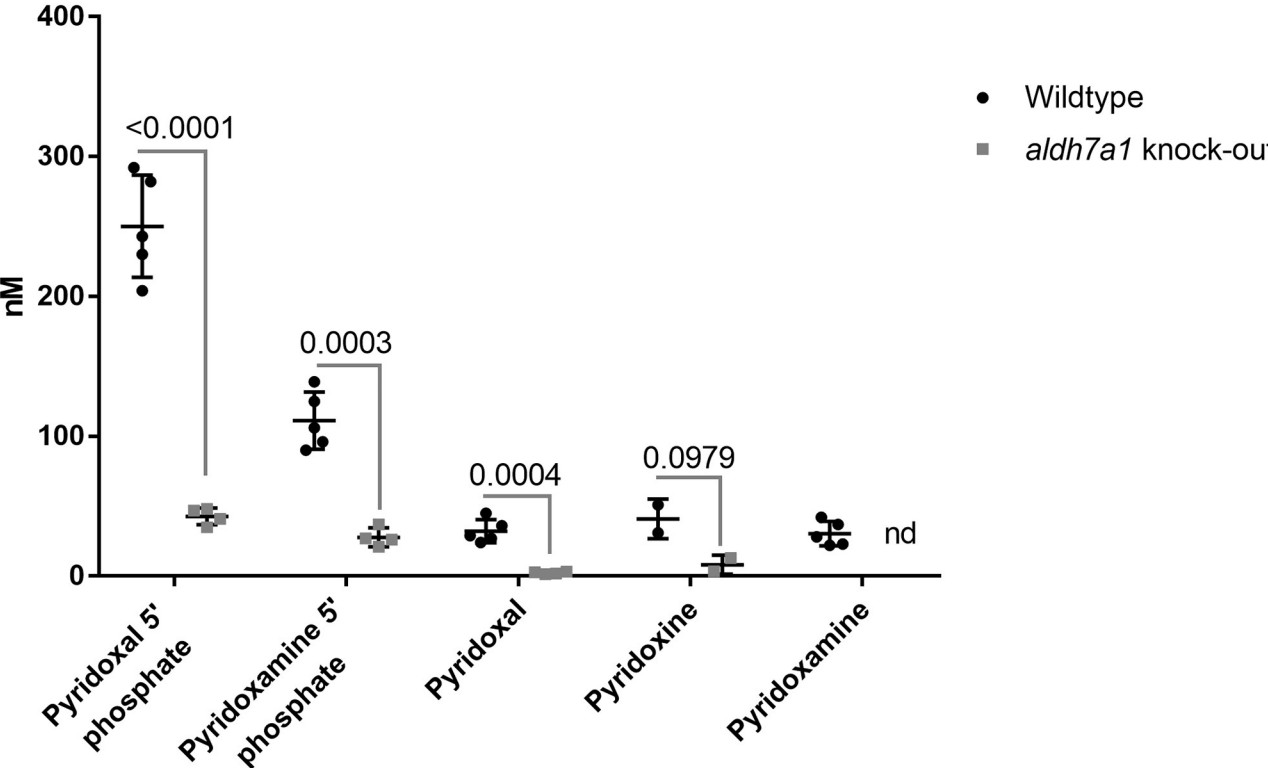

**Fig 2. Decreased levels of vitamin B6 vitamers in aldh7a1 knock-out 11 dpf embryos relative to age-matched wildtype control embryos.** Vitamin B6 vitamers levels are reduced in aldh7a1 homozygous knock-out embryos relative to the wildtype. It is shown in a scatterplot of quantification of the levels of vitamin B6 vitamers in aldh7a1 homozygous knock-out zebrafish relative to the wildtype as measured by LC-MS. The graph demonstrates decreased levels of vitamin B6 vitamers in aldh7a1 knock-out 11 dpf embryos (aldh7a1 knock-out, grey squares) relative to the age-matched wildtype control embryos (Wildtype, black circles). Each dot represents metabolite data collected from 20 embryos. For a single data point of homozygous aldh7a1 knock-out zebrafish, only 16 embryos were available, so that data point was excluded from the analysis. nd, not detectable–the levels of metabolites could not be measured. p values were calculated by unpaired student t-test for multiple measurements followed by Holm-Sadak correction for multiple analysis. Error bars, SD.

These results suggest an impairment in TCA cycle in homozygous knock-out *aldh7a1* zebrafish embryos. TCA metabolite measurement method was not optimized for zebrafish samples. All zebrafish samples were measured in one set of samples; the concentrations of the TCA metabolites in zebrafish samples can be compared to each other.

## Electron transport chain enzyme activity measurements

To investigate whether electron transport chain function was affected by the *aldh7a1* deficiency in zebrafish, we measured activities of individual electron transport chain complexes: NADH dehydrogenase (complex I), succinate dehydrogenase (complex II), cytochrome C oxidase (complex IV), ATP synthase (complex V), as well as paired complexes I+III and complexes II+III and citrate synthase. The activities of all individual, and paired complexes and citrate synthase were lower in homozygous knock-out *aldh7a1* zebrafish embryos compared to the wildtype zebrafish embryos with some differences greater than 50% relative activity (Fig 5A). All electron transport chain enzyme activity measurement results are depicted in S5C Table in S1 File. We normalized the individual and paired complexes activities to citrate synthase and depicted the results in Fig 5B. Complex II and paired complex II+III activities seemed lower than the wildtype, however complex I, IV and V activities seemed higher than the wildtype. These measurements were performed on isolated mitochondria. Therefore, these

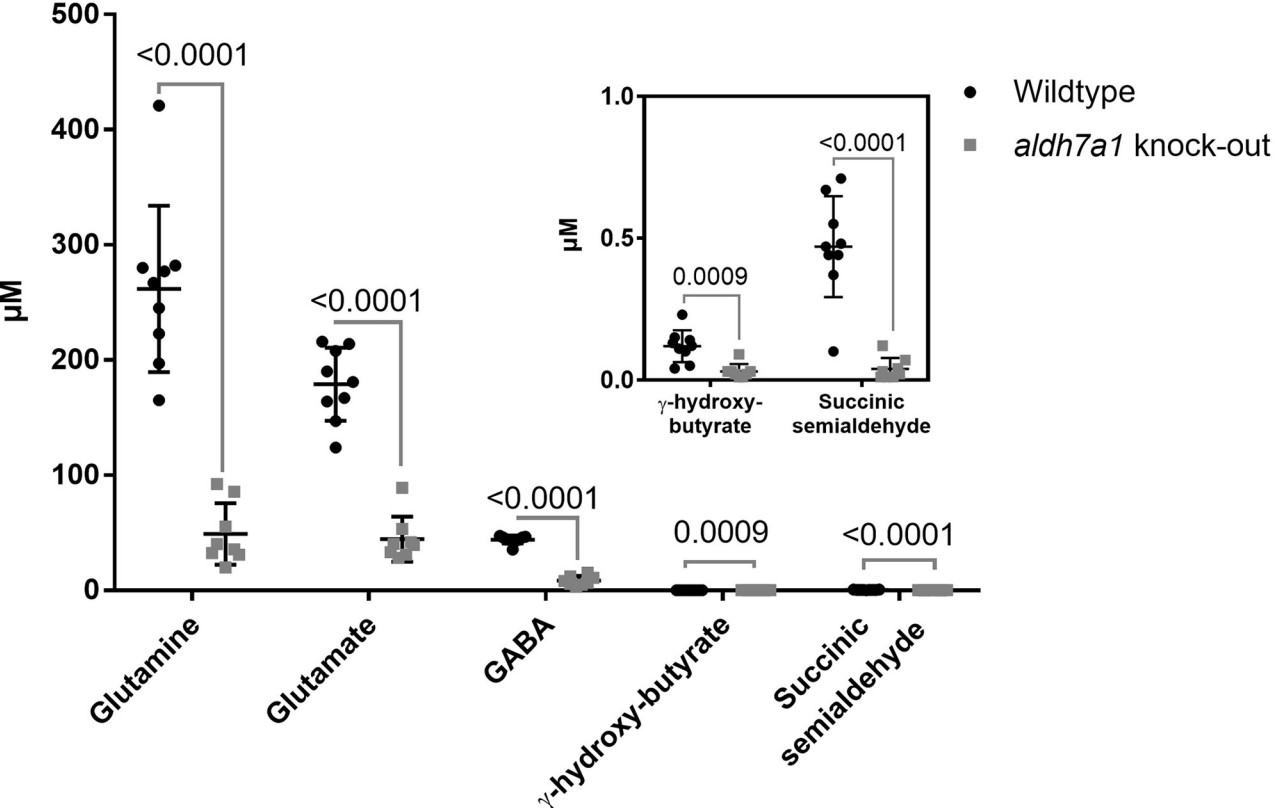

**Fig 3. Scatterplot of quantification of the levels of GABA metabolism pathway metabolites.** The levels of GABA metabolism reaction intermediates are reduced in aldh7a1 knock-out zebrafish embryos. It is shown in a scatterplot of quantification of the levels of GABA metabolism pathway metabolites, as measured by GC-MS and LC-MS/MS. The metabolite levels in 11 dpf aldh7a1 homozygous knock-out (aldh7a1 knock-out, grey squares) embryos are significantly decreased relative to the wildtype (Wildtype, black circles). Each dot represents metabolite data collected from 20 embryos. p values were calculated by unpaired student t-test for multiple measurements followed by Holm-Sadak correction for multiple analysis. Error bars, SD.

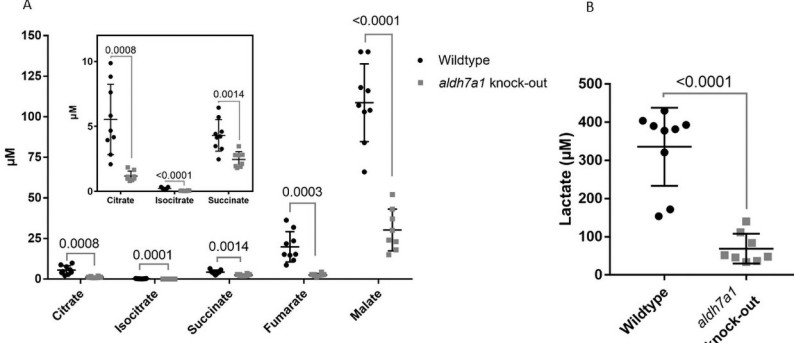

**Fig 4. A & B. Scatterplot of quantification of the TCA cycle metabolite levels.** TCA cycle metabolite levels are reduced in aldh7a1 knock-out zebrafish embryos. Scatterplot of quantification of the TCA cycle metabolite levels decreased in aldh7a1 11 dpf knock-out embryos (aldh7a1 knock-out, grey squares) relative to the wildtype age-matched controls (Wildtype, black circles), as measured by LC-MS/MS. Each dot represents metabolite data collected from 20 embryos. p values were calculated by unpaired student t-test for multiple measurements followed by Holm-Sadak correction for multiple analysis. Error bars, SD.

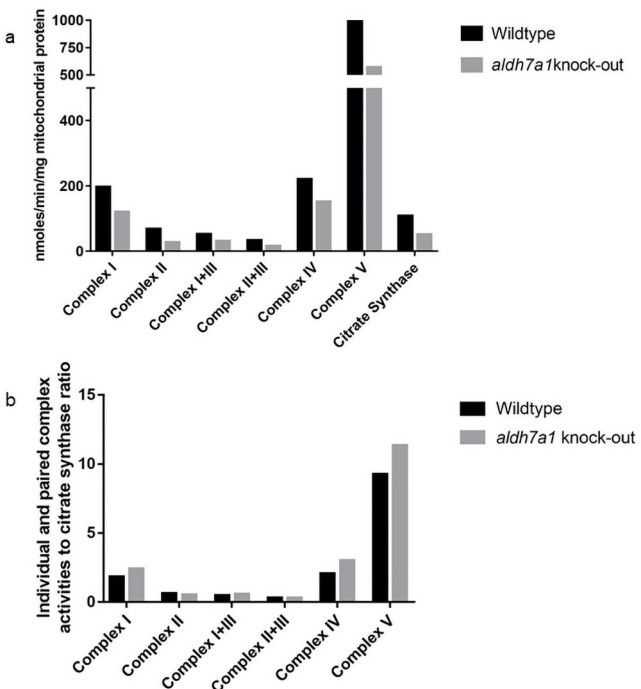

**Fig 5. A & B. Quantification of fold change in the electron transport chain enzyme activities in knock-out *aldh7a1* 11 dpf embryos relative to the wildtype age-matched controls.** Electron transport chain enzyme activities are reduced in *aldh7a1* knock-out zebrafish. 5A: individual and paired complexes and citrate synthase. 5B: ratio of individual and paired complexes to citrate synthase. Bar graph of the quantification of fold change in the electron transport chain enzyme activities in knock-out *aldh7a1* 11 dpf embryos (grey bars) relative to the wildtype age-matched controls (black bars). Each bar represents data collected from 300 embryos. The activity of each enzyme was measured in presumed homozygous 300 and 300 wildtype embryos. The values are depicted as enzyme activity in nmol/min/ mitochondrial protein mass normalized to the activity of citrate synthase (For further details please refer to Materials and Methods).

measurements should not be affected by tissue type or zebrafish homogenization, particularly when comparing against wildtype embryos isolated and measured at the same time and not between a reference tissue.

## Mitochondrial DNA qPCR

To investigate whether reduced electron transport chain enzyme activities were due to decreased mitochondrial mass, we measured mitochondrial DNA (mtDNA) content as a proxy for mitochondrial abundance. We did not observe a significant difference in mtDNA amount in homozygous knock-out *alh7a1* zebrafish embryos compared to wildtype (Fig 6).

## Discussion

We report impaired muscle electron transport chain function, and elevated CSF GABA pathway and TCA cycle metabolites in patients with PDE-*ALDH7A1*. We also report low vitamin B6 vitamers, low GABA pathway and TCA cycle metabolites and impaired electron transport chain function in knock-out *aldh7a1* zebrafish. Impaired energy production is on a spectrum ranging from decreased GABA pathway and TCA cycle metabolites and decreased electron transport chain functions in knock-out *aldh7a1* zebrafish, to elevated GABA pathway and TCA cycle metabolites in one patient with motor dysfunction and normal cognitive function on triple therapy from the infancy and impaired electron transport chain in one patient with

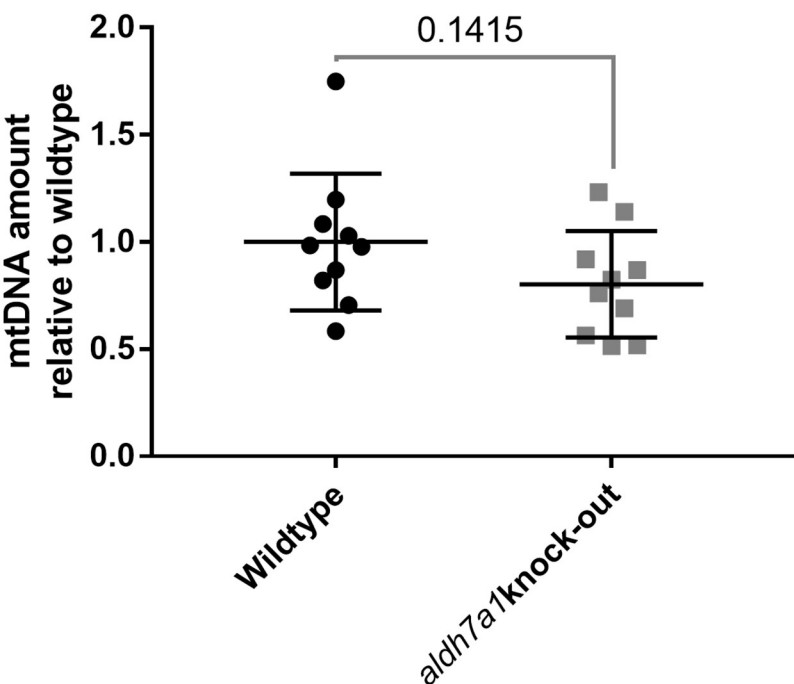

**Fig 6. mtDNA amount is not significantly reduced in aldh7a1 knock-out zebrafish.** qPCR graph of mtDNA amount from wildtype (Wildtype, black circles) and aldh7a1 homozygous knock-out (aldh7a1 knock-out, grey squares) zebrafish embryos, demonstrating only a slight reduction in mtDNA copy number in knock-out relative to the wildtype embryos. Each dot represents DNA extracted from 5 embryos. The amounts of mtDNA from knock-out and wildtype embryos were normalized to the average mtDNA amount from the wildtype embryos. p value is calculated by an unpaired two-tailed student t-test. Error bars, mean and SD.

moderate intellectual disability and seizure freedom on pyridoxine monotherapy, to normal GABA pathway and TCA cycle metabolites in a patient with normal intellectual function and seizure freedom on pyridoxine monotherapy in human PDE-*ALDH7A1*. Differences in biochemical abnormalities between knock-out *aldh7a1* zebrafish and human PDE-*ALDH7A1* are likely due to the early lethal phenotype in knock-out *aldh7a1* zebrafish representing the most severe disease phenotype and decreased acetyl-CoA supply to TCA cycle due to lysine catabolism defect secondary to α-AASA dehydrogenase deficiency and decreased succinate supply due to glutamate decarboxylase and GABA transaminase dysfunction secondary to pyridoxal-5'-phosphate deficiency resulting in impaired energy production in human PDE-*ALDH7A1*. We depicted all involved pathways and their relationship in S1 Fig in S1 File. To the best of our knowledge, we report for the first-time biomarkers of impaired energy production in human PDE-*ALDH7A1* and its animal model.

To compare our results related to energy production in PDE-*ALDH7A1*, we looked at the studies in pyridoxamine-5'-phospate oxidase (PNPO) and *PLPBP* deficiencies, which are other two vitamin B6 metabolism defects. In *PLPBP* deficiency, there was no direct role of abnormal mitochondrial metabolism in patient fibroblasts. However, *PLPBP* yeast cell (ybl036 Δ yeast cell) growth was reduced in the presence of carbon sources other than glucose, suggesting a likely effect on the mitochondrial metabolism [32]. In knock-out *pnpo* zebrafish, lactate and succinate levels were significantly decreased in TCA cycle compared to wildtype zebrafish suggesting impaired glucose utilization. Pyridoxal-5'-phosphate supplementation resulted in increased lactate and succinate levels, but not normalization [33]. In PDE-*ALDH7A1*, there is pyridoxal-5'-phosphate deficiency and neurotoxic effects of α-AASA, P6C and PA. For these

reasons, TCA cycle dysfunction is likely more prominent than PNPO and *PLPBP* deficiencies. Indeed, in our study, citrate, isocitrate, succinate, fumarate, malate, and lactate were significantly decreased in the knock-out *aldh7a1* zebrafish compared to wildtype zebrafish. These are likely due to: 1) low succinate levels due to the reduction of succinic semialdehyde production in the GABA metabolism; 2) significant reduction in lactate and oxaloacetate due to cofactor pyridoxal-5'-phosphate deficiency in the glycolysis and gluconeogenesis [14]; 3) significant reduction in citrate due to decreased production of acetyl-CoA secondary to the α-aminoadipic semialdehyde dehydrogenase deficiency in lysine catabolism and glycolysis and gluconeogenesis defects due to cofactor pyridoxal-5'-phosphate deficiency; 4) low α-ketoglutarate due to significant reduction in glutamate and succinic semialdehyde; 5) decreased intake of essential amino acids and supply of succinyl CoA due to seizures in knock-out *aldh7a1* zebrafish (smaller body size after seizure onset compared to wildtype embryos S2 Fig in S1 File). Patient 3 with PDE-*ALDH7A1* had mild elevations of citrate, succinate, malate, isocitrate and α-ketoglutarate prior to and on short-term pyridoxine therapy, similar to human mitochondrial disorders. However, patient 2 with PDE-*ALDH7A1*, who had normal intellectual functions and seizure freedom on pyridoxine monotherapy, had normal TCA cycle metabolites. This might be due to higher residual α-AASA dehydrogenase activity, lower CSF α-AASA/P6C levels and higher pyridoxal-5'-phosphate level in the central nervous system due to lower α-AASA/P6C levels. Indeed, we reported the lowest CSF α-AASA level in patient 2 previously [9]. Untreated knock-out *aldh7a1* and *pnpo* zebrafish models have decreased levels of TCA cycle metabolites compared to wildtype zebrafish, which is likely due to lethal embryonal disease onset resulting in general sickness in knock-out zebrafish models. To the best of our knowledge, we report for the first time TCA cycle abnormalities in the knock-out *aldh7a1* zebrafish and human PDE-*ALDH7A1*.

To compare our results related to GABA metabolism, we looked at studies in PNPO and *PLPBP* deficiencies. Untreated knock-out *pnpo* zebrafish had low glutamate (not significant) and low GABA levels compared to wildtype zebrafish [33]. Our untreated knock-out *aldh7a1* zebrafish had low levels of GABA pathway metabolites compared to wildtype zebrafish. It seems that both zebrafish models have similar biochemical features of GABA pathway metabolism which is likely due to lethal embryonal disease onset resulting in general sickness in knock-out zebrafish models. Pre-treatment CSF metabolomic analysis of patients with *PLPBP* deficiency showed mild elevation of α-ketoglutarate [32]. Markedly elevated glutamate and α-ketoglutarate levels in CSF prior to pyridoxine therapy did not normalize on short-term pyridoxine therapy in patient 3 with PDE-*ALDH7A1*. Whereas there was normal glutamate and α-ketoglutarate levels in CSF on long-term pyridoxine therapy in patient 2 with likely milder PDE-*ALDH7A1* in our study. Elevated glutamate is likely due to ongoing glutamate decarboxylase and GABA transaminase deficiencies secondary to pyridoxal-5'-phosphate deficiency in human PDE-*ALDH7A1*, as the accumulation of CSF α-AASA and PA do not normalize in patients with PDE-*ALDH7A1* despite treatments [12, 17]. Possible explanations for the differences of CSF glutamate level between patient 2 and patient 3 are multifold and include that patient 3 likely has lower residual α-AASA dehydrogenase activity, lower pyridoxal-5'-phosphate level and higher CSF α-AASA/P6C levels in the central nervous system. Indeed, we previously reported a higher CSF α-AASA level in patient 3 which likely supports our possible explanations [9]. Glutamate is the major excitatory neurotransmitter in the central nervous system. Increased glutamate levels in the extracellular space results in neuronal cell death. It is important to keep CSF glutamate levels at the physiological level of 10 μM [2, 34–36]. Elevated glutamate levels in the central nervous system are likely one of the contributing factors to the disease severity in PDE-*ALDH7A1*. We also report significantly low levels of GABA, glutamine, glutamate, γ-hydroxy-butyrate, and succinic semialdehyde in the GABA metabolism in

our knock-out *aldh7a1* zebrafish, which is likely severe spectrum of the disease resulting in death at embryonic life in zebrafish. To the best of our knowledge, this is the first study to report glutamate toxicity in the central nervous system as a possible contributor of the disease severity in human PDE-*ALDH7A1*.

We report decreased electron transport chain complex activities in human PDE-*ALDH7A1* and in knock-out *aldh7a1* zebrafish. The reasons of decreased electron transport chain activities can be multifold: 1) decreased conversion of nicotinamide adenine dinucleotide (NAD$^+$) into three equivalents of reduced NAD$^+$ (NADH) and flavin adenine dinucleotide (FAD) into one equivalent of FADH$_2$ in TCA cycle and both are used in electron transport chain; 2) decreased succinate, a substrate for complex II in the electron transport chain; 3) decreased δ-aminolevulinic acid synthase 1 and mitochondrial cysteine desulfurase due to pyridoxal-5'-phosphate deficiency, both participate in iron heme synthesis and iron-sulfur cluster synthesis [14]. Iron-sulfur clusters are the core redox centres of electron transport chain complexes I, II and III. Heme are present in complex II, III and IV as protein-bound prosthetic groups [14, 37]; 4) Citrate synthase is the first enzyme in the TCA cycle that catalyzes the conversion of acetyl-CoA into citrate. All these may contribute to electron transport chain dysfunction and decreased ATP production in human PDE-*ALDH7A1* and in knock-out *aldh7a1* zebrafish. We report mitochondrial dysfunction in human PDE-*ALDH7A1* and in the knock-out *aldh7a1* zebrafish which may possibly contribute to disease severity.

The knock-out *pnpo* zebrafish model showed low pyridoxal, and high pyridoxamine and pyridoxamine-5'-phosphate levels, however pyridoxal-5'-phosphate levels were similar to wild-type [33]. The knock-out *plpbp* zebrafish had significantly low levels of pyridoxal-5'-phosphate and pyridoxal, but low levels were not significant for pyridoxamine-5'-phosphate and pyridoxine levels [32, 33]. Previously reported knock-out *aldh7a1* zebrafish also showed a statistically significant decrease in pyridoxal and pyridoxamine-5'-phosphate, but there was no statistically significant difference in pyridoxal-5'-phosphate levels compared to wildtype [38]. In our study, vitamin B6 vitamers including pyridoxal-5'-phosphate, pyridoxamine-5'-phosphate, pyridoxal, and pyridoxamine levels were significantly low in homozygous knock-out *alh7a1* zebrafish embryos compared to wildtype. It is likely that depletion of pyridoxal-5'-phosphate consumes other forms of vitamin B6 vitamers including pyridoxamine, pyridoxal and pyridoxamine-5'-phosphate in an effort to maintain cellular function.

Our study has several limitations: 1) single PDE-*ALDH7A1* patient with electron transport chain activity measurements (no remaining CSF sample for the measurement of CSF GABA pathway and TCA cycle metabolites), and single patient with PDE-*ALDH7A1* (pre-treatment and short-term pyridoxine treatment) and single patient with PDE-*ALDH7A1* (long term pyridoxine treatment) patient to measure CSF GABA pathway and TCA cycle metabolites. Our study reviewed 12 patients with PDE-*ALDH7A1* from a single neurometabolic clinic at The Hospital for Sick Children. Additionally, we received information for eight patients with PDE-*ALDH7A1* from CIMDRN and four different metabolic genetic centers in Canada. None of those patients had muscle electron transport chain enzyme activity measurements or remaining CSF sample for us to include into this study. We also did not include information of those eight additional patients with PDE-*ALDH7A1* from other Canadian centers, as our study did not aim to report all patients diagnosed with PDE-*ALDH7A1* in Canada. It seems that muscle biopsy and lumbar puncture are invasive investigations and not routinely performed in patients for the diagnosis of PDE-*ALDH7A1*. Confirmation of the diagnosis of PDE-*ALDH7A1* is mostly by clinical suspicion, measurement of disease specific biomarkers and/or molecular genetic investigations. Unfortunately, we did not have additional patient samples with PDE-*ALDH7A1* to include into our study cohort, which would have supported our concept better. Multicenter studies will require multiple institutional research ethics board

applications, funding, consenting patients and data collection. We did not have funding support to arrange this scale study at this time. The international PDE registry (https://pdeonline. org/pderegistry.html) has registered over 130 patients between 2014 and 2021, which is a worldwide registry. We think that despite having information for a smaller number of patients in this study compared to PDE Registry, it can still be considered as a good representation of a larger patient population for a rare neurometabolic disease. 2) due to the requirements to use large number of zebrafish embryos for metabolite measurements, we were not able to use wild-type siblings of knock-out homozygous *aldh7a1* zebrafish or include more enzymes to assess integrity of metabolite and electron transport chain activity measurements to protect animal welfare and to use the minimum number of animals; 3) we were not able to use siblings of knock-out homozygous *aldh7a1* zebrafish due to the requirements to genotype each embryo for the confirmation of homozygosity, as this required more funding, high resolution melting machine time for genotyping and technician time to perform the genotyping; 4) we also did not include heterozygous siblings due to the above reasons. However, we would expect that heterozygous siblings would have had similar metabolite profiles to wildtype and serve as a control group. Indeed, we previously reported that heterozygous *aldh7a1* zebrafish had similar levels of α-AASA, P6C and PA compared to wildtype [13]; 5) we cannot exclude the possibility of generalized sickness reflecting our results suggestive of impaired energy production in knock-out *aldh7a1* zebrafish. Despite these limitations, we think that our study is an important study to report a possible central nervous system glutamate toxicity and impaired energy production in the neuropathogenesis of PDE-*ALDH7A1* as potential contributory factors to disease severity.

## Conclusion

We report biomarkers of impaired energy production and central nervous system glutamate accumulation in human PDE-*ALDH7A1* and defects in GABA pathway, TCA cycle and electron transport chain in knock-out *aldh7a1* zebrafish. Block in the lysine catabolism due to α-AASA dehydrogenase deficiency and abnormal GABA pathway due to pyridoxal-5'-phosphate deficiency likely result in abnormal functions of TCA cycle and electron transport chain and contribute to impairment of energy production. These may contribute to the neuronal cell death during the early stages of brain development in addition to α-AASA, P6C and PA neurotoxicity and pyridoxal-5'-phosphate deficiency in the central nervous system. We think that our study population is a good representation of a larger patient population with PDE-*ALDH7A1*, which consisted 15% of the patients enrolled into the International PDE Registry (https://pdeonline.org/pderegistry.html) study. More patients with range of disease severity are required to investigate GABA pathway and TCA cycle metabolism and electron transport chain at different time points to shed lights on the neuropathogenesis of PDE-*ALDH7A1*. These are invasive investigations and require multicenter institutional research ethics board approvals and patient and parent participation as well as funding support.

## Supporting information

**S1 File.**
(PDF)

## Acknowledgments

As principal author of this study, I, Saadet Mercimek-Andrews, would like to thank Dr. Berge Minassian for his tremendous support by providing laboratory space for the *aldh7a1* zebrafish

studies, encouragement for the development of my research program, and his ongoing, excellent guidance and mentorship. We would like to thank Dr. Jim Dowling for his guidance on the zebrafish model development. We would like to thank the Institutional Zebrafish Facility, Ramil Noche, Xiucheng Cui, Nikita Zabinyakov, Yulia Kungurova, and Li Qing She for maintaining the model, and preparing the zebrafish samples for metabolite measurements. We would also like to thank Valeriy Levandovskiy for the measurement of electron transport chain enzyme activities. We would like to thank Dr. Mirjam M.M. Wamelink for the management of CSF GABA pathway and TCA cycle metabolite measurements, as well as for her critical review of the manuscript.

## Author Contributions

**Conceptualization:** Saadet Mercimek-Andrews.

**Data curation:** Anastasia Minenkova, Saadet Mercimek-Andrews.

**Formal analysis:** Jessie Cameron, Rob Barto.

**Funding acquisition:** Saadet Mercimek-Andrews.

**Investigation:** Erwin E. W. Jansen, Thomas Hurd, Lauren MacNeil, Gajja S. Salomons, Saadet Mercimek-Andrews.

**Methodology:** Saadet Mercimek-Andrews.

**Project administration:** Saadet Mercimek-Andrews.

**Resources:** Anastasia Minenkova, Erwin E. W. Jansen, Jessie Cameron, Rob Barto, Thomas Hurd, Lauren MacNeil, Gajja S. Salomons.

**Supervision:** Saadet Mercimek-Andrews.

**Validation:** Anastasia Minenkova, Erwin E. W. Jansen, Rob Barto, Thomas Hurd, Saadet Mercimek-Andrews.

**Visualization:** Saadet Mercimek-Andrews.

**Writing – original draft:** Anastasia Minenkova, Saadet Mercimek-Andrews.

**Writing – review & editing:** Anastasia Minenkova, Erwin E. W. Jansen, Jessie Cameron, Rob Barto, Thomas Hurd, Lauren MacNeil, Gajja S. Salomons, Saadet Mercimek-Andrews.

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
