## [Decision Letter · Decision Letter 0]

7 Apr 2021

PONE-D-20-40994

Is impaired energy production a novel insight into the pathogenesis of pyridoxine-dependent epilepsy due to biallelic variants in ALDH7A1?

PLOS ONE

Dear Dr. Mercimek-Andrews,

Thank you for submitting your manuscript to PLOS ONE. After careful consideration, we feel that it has merit but does not fully meet PLOS ONE’s publication criteria as it currently stands. Therefore, we invite you to submit a revised version of the manuscript that addresses the points raised during the review process.

In your revised manuscript please try to address the cogent and constructive comments of the two reviewers as fully as possible.

We look forward to receiving your revised manuscript.

Kind regards,

Israel Silman

Academic Editor

PLOS ONE

Journal Requirements:

3. Thank you for including your ethics statement: "The study was approved by Institutional Research Ethics Board (REB#1000050808) for human participants."   

4. In your Methods section, please provide additional information about the participant recruitment method and the demographic details of your participants. Please ensure you have provided sufficient details to replicate the analyses such as: a) the recruitment date range (month and year), b) a description of any inclusion/exclusion criteria that were applied to participant recruitment, c) a table of relevant demographic details, d) a statement as to whether your sample can be considered representative of a larger population, e) a description of how participants were recruited, and f) descriptions of where participants were recruited and where the research took place.

5. Please provide additional details regarding participant consent. In the ethics statement in the Methods and online submission information, please ensure that you have specified (1) whether consent was informed and (2) what type you obtained (for instance, written or verbal, and if verbal, how it was documented and witnessed). If your study included minors, state whether you obtained consent from parents or guardians. If the need for consent was waived by the ethics committee, please include this information.

6. In the ethics statement in the manuscript and in the online submission form, please provide additional information about the patient records/samples used in your retrospective study, including the source of the medical records/samples analyzed in this work (e.g. hospital, institution or medical center name).

7. At this time, we request that you  please report additional details in your Methods section regarding animal care, as per our editorial guidelines:

(1) Please state the source of the zebrafish

(2) Please include the method of euthanasia

(3) Please describe the care received by the animals, including the frequency of monitoring and the criteria used to assess animal health and well-being, and any steps taken to minimise animal suffering.

Thank you for your attention to these requests.

Reviewers' comments:

Reviewer's Responses to Questions

**Comments to the Author**

1. Is the manuscript technically sound, and do the data support the conclusions?

Reviewer #1: Partly

Reviewer #2: Partly

2. Has the statistical analysis been performed appropriately and rigorously? 

Reviewer #1: Yes

Reviewer #2: Yes

3. Have the authors made all data underlying the findings in their manuscript fully available?

Reviewer #1: Yes

Reviewer #2: No

4. Is the manuscript presented in an intelligible fashion and written in standard English?

Reviewer #1: Yes

Reviewer #2: Yes

5. Review Comments to the Author

Reviewer #1: In this paper, the authors described three previously reported patients diagnosed with PDE-ALDH7A1, and measured GABA pathway, tricarboxylic acid cycle metabolites, electron transport chain activities, and vitamin B6 vitamers in patients with PDE-ALDH7A1 and in knock-out aldh7a1 zebrafish model that have also been reported by themselves before. Besides, the author further pointed out that central nervous system glutamate toxicity and impaired energy production may play important roles in the disease neuropathogenesis and severity in human PDE-ALDH7A1. However, for the metabolites of patients, the limited sample size and variety of treatments for the patients make the results inconclusive. The major and minor comments related to the paper are described below.

Major comments

Introduction

The authors could elaborate on the number of reported cases and then state what metabolic tests those patients had, which would make it easier to understand.

Materials and Methods section

Patients

The authors have noted that “All patients were previously reported for their phenotypes, genotypes and treatment outcomes”. To aid in transparency, it would be helpful if the authors documented all of the manuscripts where a specific patient has been published.

Result

Patients

Although All patients were previously reported for their phenotypes, genotypes and treatment outcomes, these were not clear in this study.

—Follow-up of these three patients can be described, such as oral pyridoxine dose, seizure outcome, and neurodevelopment.

—Patient 2 was also treated with lysine restriction and arginine supplementation, which should be mentioned in this paper, and whether this treatment had any effect on the detection of GABA and TCA metabolites.

—The genetic age of patient 3 is unknown.

—In the result section of patient 3, the results of CSF GABA pathway and TCA cycle metabolite results prior to pyridoxine therapy were compared with the age-matched group, but the age-matched group only has one column. So, the age of the control group was 2 weeks or 7 months? Because there are several comparisons in the text.

Discussion

—In the discussion part, the results of GABA pathway, tricarboxylic acid cycle metabolites, electron transport chain activities and vitamin B6 vitamers were discussed. However, although the authors carried on the substantial explanation to the meaningful result, they should emphasize the important points, not just the list of possibilities.

—The authors classified the three reported patients into severe and mild phenotypes. It would aid the reader if the authors stated how they defined a severe and mild phenotype. And it should be stated which patients showed severe phenotype and which showed mild.

—The levels of TCA cycle metabolites in patient 2 were not different from those in the age control group, which was not discussed in the discussion section.

—Seizures in PDE are known to be caused by elevated glutamate levels and reduced GABA levels. The seizures in patients 2 and 3 were controlled, but the metabolic results showed that glutamate levels were normal in patient 2 and increased in patient 3 compared to age matched controls. This result requires additional explanation.

—In the untreated knock-out PDE zebrafish model, the levels of GABA-related metabolites, especially glutamate, are reduced, but the levels of glutamate were increased in patients with severe phenotypes, which also requires further explanation and explanation. A similar situation occurs in the levels of TCA metabolites.

—The authors examined and compared GABA pathway metabolites and TCA cycle metabolites, electron transport chain activities and vitamin B6 vitamers in homozygous knock-out aldh7a1 zebrafish and wild-type. If so, could heterozygous zebrafish also serve as a control group.

Minor comments

—Paragraph 12, line 251-253

“He had elevated urine α-AASA (8.8 mmol/mol creatinine; reference range 0-0.5 and compound heterozygous variants (c.500A>G (p. Asn167Ser) and c.1481+1G>T) in ALDH7A1 confirming the diagnosis of PDE-ALDH7A1 at the age of 11 years (9).” A parenthesis is missing after 0.5.

Reviewer #2: I received the article entitled “Is impaired energy production a novel insight into the pathogenesis of pyridoxine 2 dependent epilepsy due to biallelic variants in ALDH7A1?” by Minenkova et al

Overall, I found this to be an interesting paper,- but have some comments/concerns:

The authors need to include, under the materials and methods section, the deuterated internal standards and where they were procured from, needs to be included, etc.

A representative chromatogram showing all of the analytes analyzed would be nice; can include in supplementary materials section

Could the authors include more information about the method used to analyze samples

What column, what flow-rate, what are the retention times of the various components

How much sample was injected, was a column guard used ?

What sort of linear range was observed - see line 156

The authors need to denote that they were using ESI negative ion mode and also include other parameters such as CE (collision energy), DP, CXP, temperature, etc

What linear range did the authors receive for the various analyst that were analyzed

Might want to consider a Table which can be added to Supp, Materials section

The authors need to provide sufficient information for someone to reproduce the reported work

The authors discuss the metabolite AASA but don't include AASA/P6C concentrations.

Also, it is not AASA but rather PC6H which is metabolized to 6-Oxo-Pip

6-0x0-pip concentrations were not analyzed in this work, which detracts from the paper

My major concern is the lack of patient samples

Obviously it is difficult to get ahold of samples from PDE patients, they are precious

For example, not all of the patients had the same metabolic disturbances. A few of those metabolites were outside of controls but not significantly so. Certainly subject 3 had very elevated glutamate but that was only one patient.

I really like what authors are trying to propose, a couple of additional patient samples to support the concept would be wise

6. PLOS authors have the option to publish the peer review history of their article (what does this mean?). If published, this will include your full peer review and any attached files.

Reviewer #1: No

Reviewer #2: No

---

## [Author Response · Author response to Decision Letter 0]

12 Jul 2021

Response to Reviewers and Editorial Board

1. Answer: Thanks very much. We added this information into the cover letter. We hope that our responses are satisfactory to the Editorial Board. 

2. Guidelines for resubmitting your figure files are available below the reviewer comments at the end of this letter. While revising your submission, please upload your figure files to the Preflight Analysis and Conversion Engine (PACE) digital diagnostic tool, https://pacev2.apexcovantage.com/. PACE helps ensure that figures meet PLOS requirements. To use PACE, you must first register as a user. Registration is free. Then, login and navigate to the UPLOAD tab, where you will find detailed instructions on how to use the tool. If you encounter any issues or have any questions when using PACE, please email PLOS at figures@plos.org. Please note that Supporting Information files do not need this step.

Answer: Thanks very much for this information. We applied these. We hope that our responses are satisfactory to the Editorial Board. 

Answer: Thanks very much for this information. This is not applicable for our study. We hope that our responses are satisfactory to the Editorial Board. 

Answer: All data was included for metabolite measurements as supplemental tables during revisions process. There is no other data available. We hope that our responses are satisfactory to the Editorial Board. 

Answer: Thanks very much for this information. This is not applicable for our study. We hope that our responses are satisfactory to the Editorial Board. 

Answer: Thanks very much for this question. This information is included as supplemental data. We hope that our responses are satisfactory to the Editorial Board. 

6. Thank you for including your ethics statement: "The study was approved by Institutional Research Ethics Board (REB#1000050808) for human participants." 

Answer: Thanks very much for this question. This information is included. We hope that our responses are satisfactory to the Editorial Board. 

Answer: Thanks very much for this question. This information is included. We hope that our responses are satisfactory to the Editorial Board. 

7. In your Methods section, please provide additional information about the participant recruitment method and the demographic details of your participants. Please ensure you have provided sufficient details to replicate the analyses such as: a) the recruitment date range (month and year), b) a description of any inclusion/exclusion criteria that were applied to participant recruitment, c) a table of relevant demographic details, d) a statement as to whether your sample can be considered representative of a larger population, e) a description of how participants were recruited, and f) descriptions of where participants were recruited and where the research took place.

Answer: Thanks very much for this question. This information is included into the methods and discussion sections. All information is highlighted yellow. We hope that our responses are satisfactory to the Editorial Board. 

8. Please provide additional details regarding participant consent. In the ethics statement in the Methods and online submission information, please ensure that you have specified (1) whether consent was informed and (2) what type you obtained (for instance, written or verbal, and if verbal, how it was documented and witnessed). If your study included minors, state whether you obtained consent from parents or guardians. If the need for consent was waived by the ethics committee, please include this information.

Answer: Thanks very much for these questions. This information is included into the methods and all information is highlighted yellow. We hope that our responses are satisfactory to the Editorial Board. 

9. In the ethics statement in the manuscript and in the online submission form, please provide additional information about the patient records/samples used in your retrospective study, including the source of the medical records/samples analyzed in this work (e.g. hospital, institution or medical center name).

Answer: Thanks very much for these questions. This information is included into the methods and all information is highlighted yellow. We hope that our responses are satisfactory to the Editorial Board. 

10. At this time, we request that you please report additional details in your Methods section regarding animal care, as per our editorial guidelines:

(1) Please state the source of the zebrafish

(2) Please include the method of euthanasia

(3) Please describe the care received by the animals, including the frequency of monitoring and the criteria used to assess animal health and well-being, and any steps taken to minimise animal suffering.

Answer: Thanks very much for these questions. This information is included into the methods and all information is highlighted yellow. We hope that our responses are satisfactory to the Editorial Board. 

 

Reviewer #1: 

Major comments

Introduction

The authors could elaborate on the number of reported cases and then state what metabolic tests those patients had, which would make it easier to understand.

Answer: Thanks very much for this question. We included these into the introduction and highlighted yellow. We hope that our responses would be satisfactory to Reviewer 1. 

Materials and Methods section

Patients

The authors have noted that “All patients were previously reported for their phenotypes, genotypes and treatment outcomes”. To aid in transparency, it would be helpful if the authors documented all of the manuscripts where a specific patient has been published.

Answer: Thanks very much for this question. We included these into the Material and Methods, Patients and in the results and highlighted yellow. We hope that our responses would be satisfactory to Reviewer 1.

Result

Patients

—Follow-up of these three patients can be described, such as oral pyridoxine dose, seizure outcome, and neurodevelopment.

Answer: Thanks very much for this question. We included these into the results and highlighted yellow. We hope that our responses would be satisfactory to Reviewer 1.

—Patient 2 was also treated with lysine restriction and arginine supplementation, which should be mentioned in this paper, and whether this treatment had any effect on the detection of GABA and TCA metabolites.

Answer: Thanks very much for this question. The CSF samples were not collected on arginine and lysine-restricted diet, only on pyridoxine monotherapy. This is specified in results and highlighted yellow. We hope that our responses would be satisfactory to Reviewer 1.

—The genetic age of patient 3 is unknown.

Answer: Thanks very much for this question. The information is included and highlighted yellow. We hope that our responses would be satisfactory to Reviewer 1.

—In the result section of patient 3, the results of CSF GABA pathway and TCA cycle metabolite results prior to pyridoxine therapy were compared with the age-matched group, but the age-matched group only has one column. So, the age of the control group was 2 weeks or 7 months? Because there are several comparisons in the text.

Answer: Thanks very much for this question. The information is included and highlighted yellow for each metabolite in the Table 1 and 2 and age range for each metabolite was entered under the table for Table 2 and in the table for Table 1. Indeed, in Table 1 for patient 3, the reference ranges were same for prior to pyridoxine therapy and on pyridoxine therapy. We hope that our responses would be satisfactory to Reviewer 1.

Discussion

—In the discussion part, the results of GABA pathway, tricarboxylic acid cycle metabolites, electron transport chain activities and vitamin B6 vitamers were discussed. However, although the authors carried on the substantial explanation to the meaningful result, they should emphasize the important points, not just the list of possibilities.

Answer Thanks very much for this question. The information is included into the conclusion and highlighted yellow. We hope that our responses would be satisfactory to Reviewer 1.

—The authors classified the three reported patients into severe and mild phenotypes. It would aid the reader if the authors stated how they defined a severe and mild phenotype. And it should be stated which patients showed severe phenotype and which showed mild.

Answer: Thanks very much for this question. As there was no objective measurement of disease severity, we listed the specifications for each patient for their TCA, electron transport and GABA pathways. We also updated information in the abstract. All changes are highlighted yellow. We hope that our responses would be satisfactory to Reviewer 1.

—The levels of TCA cycle metabolites in patient 2 were not different from those in the age control group, which was not discussed in the discussion section.

Answer: Thanks very much for this question. We included information into the discussion. All changes are highlighted yellow. We hope that our responses would be satisfactory to Reviewer 1.

—Seizures in PDE are known to be caused by elevated glutamate levels and reduced GABA levels. The seizures in patients 2 and 3 were controlled, but the metabolic results showed that glutamate levels were normal in patient 2 and increased in patient 3 compared to age matched controls. This result requires additional explanation.

Answer: Thanks very much for this question. We included information into the discussion. All changes are highlighted yellow. We hope that our responses would be satisfactory to Reviewer 1

—In the untreated knock-out PDE zebrafish model, the levels of GABA-related metabolites, especially glutamate, are reduced, but the levels of glutamate were increased in patients with severe phenotypes, which also requires further explanation and explanation. A similar situation occurs in the levels of TCA metabolites.

Answer: Thanks very much for this question. We included this information into the discussion. All changes are highlighted yellow. We hope that our responses would be satisfactory to Reviewer 1

—The authors examined and compared GABA pathway metabolites and TCA cycle metabolites, electron transport chain activities and vitamin B6 vitamers in homozygous knock-out aldh7a1 zebrafish and wild-type. If so, could heterozygous zebrafish also serve as a control group.

Answer: Thanks very much for this question. We included this information into the discussion. All changes are highlighted yellow. We hope that our responses would be satisfactory to Reviewer 1

Minor comments

—Paragraph 12, line 251-253

“He had elevated urine α-AASA (8.8 mmol/mol creatinine; reference range 0-0.5 and compound heterozygous variants (c.500A>G (p. Asn167Ser) and c.1481+1G>T) in ALDH7A1 confirming the diagnosis of PDE-ALDH7A1 at the age of 11 years (9).” A parenthesis is missing after 0.5.

Answer: Thanks very much for this correction, applied and highlighted. We hope that our responses would be satisfactory to Reviewer 1

 

Reviewer #2: 

1. The authors need to include, under the materials and methods section, the deuterated internal standards and where they were procured from, needs to be included, etc.

Answer: Thanks very much for these questions. All these information is included into the material and methods and highlighted yellow. We hope that our responses would be satisfactory to Reviewer 2.

2. A representative chromatogram showing all of the analytes analyzed would be nice; can include in supplementary materials section

Answer: Thanks very much for this question. Unfortunately, we did not have any figure to submit. We hope that our responses would be satisfactory to Reviewer 2.

3. Could the authors include more information about the method used to analyze samples

What column, what flow-rate, what are the retention times of the various components

How much sample was injected, was a column guard used?

What sort of linear range was observed - see line 156

Answer: Thanks very much for these questions. All these information is included into the material and methods and highlighted yellow. We hope that our responses would be satisfactory to Reviewer 2.

4. The authors need to denote that they were using ESI negative ion mode and also include other parameters such as CE (collision energy), DP, CXP, temperature, etc

What linear range did the authors receive for the various analyst that were analyzed

Might want to consider a Table which can be added to Supp, Materials section

The authors need to provide sufficient information for someone to reproduce the reported work

Answer: Thanks very much for these questions. All these information is included into the material and methods and highlighted yellow. We hope that our responses would be satisfactory to Reviewer 2.

5. The authors discuss the metabolite AASA but don't include AASA/P6C concentrations.

Also, it is not AASA but rather PC6H which is metabolized to 6-Oxo-Pip

6-0x0-pip concentrations were not analyzed in this work, which detracts from the paper

Answer: Thanks very much for this comment. We previously showed biochemical abnormalities in our model and reported in 2017 (PMID: 29053735). In our current study, we confirmed that all seizing embryos were homozygous by genotyping. Genotyping was chosen as quick, high throughput and cost-effective way to apply in our research study. If we would have chosen to measure biochemical parameters additionally, we would have been using more embryos. According to our ethical responsibilities for animal welfare, scientific, legal and economic reasons, we made sure to use minimum number of animals in our study. We also did not apply biochemical confirmation as an additional method due to associated high costs. We hope that our response would be satisfactory to Reviewer 2. 

6. My major concern is the lack of patient samples

Obviously it is difficult to get ahold of samples from PDE patients, they are precious

Answer: Thanks very much for this comment. We agree with the Reviewer 2 that more patients would have been better to support our study findings. Unfortunately, we have not identified any more patients and explained our extensive efforts for the patient cohort that we worked to find more patient in the material and methods and in the discussion. We hope that our response would be satisfactory to Reviewer 2. 

7. For example, not all of the patients had the same metabolic disturbances. A few of those metabolites were outside of controls but not significantly so. Certainly subject 3 had very elevated glutamate but that was only one patient.

Answer: Thanks very much for this comment. We agree with the Reviewer 2 that more patients would have been better to support our study findings. Unfortunately, we have not identified any more patients and explained our extensive efforts for the patient cohort that we worked to find more patient in the material and methods and in the discussion. We hope that our response would be satisfactory to Reviewer 2. 

8. I really like what authors are trying to propose, a couple of additional patient samples to support the concept would be wise. 

Answer: Thanks very much for this comment. We agree with the Reviewer 2 that more patients would have been better to support our study findings. We think that a multicenter follow-up study would be great to initiate, however this would require grant applications and funding support. Unfortunately, we have not identified any more patients and explained our extensive efforts for the patient cohort that we worked to find more patient in the material and methods and in the discussion. We hope that our response would be satisfactory to Reviewer 2.

---

## [Decision Letter · Decision Letter 1]

27 Jul 2021

PONE-D-20-40994R1

Is impaired energy production a novel insight into the pathogenesis of pyridoxine-dependent epilepsy due to biallelic variants in ALDH7A1?

PLOS ONE

Dear Dr. Mercimek-Andrews,

Thank you for submitting your manuscript to PLOS ONE. After careful consideration, we feel that it has merit but does not fully meet PLOS ONE’s publication criteria as it currently stands. Therefore, we invite you to submit a revised version of the manuscript that addresses the points raised during the review process.

In your revised version, please address the minor comments of the two reviewers.

We look forward to receiving your revised manuscript.

Kind regards,

Israel Silman

Academic Editor

PLOS ONE

Journal Requirements:

Reviewers' comments:

Reviewer's Responses to Questions

**Comments to the Author**

1. If the authors have adequately addressed your comments raised in a previous round of review and you feel that this manuscript is now acceptable for publication, you may indicate that here to bypass the “Comments to the Author” section, enter your conflict of interest statement in the “Confidential to Editor” section, and submit your "Accept" recommendation.

Reviewer #1: All comments have been addressed

Reviewer #2: All comments have been addressed

2. Is the manuscript technically sound, and do the data support the conclusions?

Reviewer #1: (No Response)

Reviewer #2: Yes

3. Has the statistical analysis been performed appropriately and rigorously? 

Reviewer #1: (No Response)

Reviewer #2: Yes

4. Have the authors made all data underlying the findings in their manuscript fully available?

Reviewer #1: (No Response)

Reviewer #2: Yes

5. Is the manuscript presented in an intelligible fashion and written in standard English?

Reviewer #1: (No Response)

Reviewer #2: Yes

6. Review Comments to the Author

Reviewer #1: In Introduction section, the authors described “Less than 200 patients with PDE-ALDH7A1 were reported in the literature (5-8).” However, the number of previously reported cases with ALDH7A1 mutations summarized in this paper was biased. In the latest literature, Jiao X, et al. described the number of cases currently reported carrying mutations in the ALDH7A1, which is far greater than described in this paper. (Jiao X, Gong P, Wu Y, Zhang Y, Yang Z. Analysis of the Phenotypic Variability as Well as Impact of Early Diagnosis and Treatment in Six Affected Families With ALDH7A1 Deficiency. Front Genet. 2021;12:644447, and Jiao X, Xue J, Gong P, Wu Y, Zhang Y, Jiang Y, Yang Z. Clinical and genetic features in pyridoxine-dependent epilepsy: a Chinese cohort study. Dev Med Child Neurol. 2020;62(3):315-321.).

Reviewer #2: I felt that the authors have attempted to address concerns which were raised.

However, I do think it is important to point out to your readers that AASA is really not as the aldehyde, but rather the intramolecular cyclized form; J Inherit Metab Dis 2019 May;42(3):565-574. doi: 10.1002/jimd.12059. Epub 2019 Mar 11.

7. PLOS authors have the option to publish the peer review history of their article (what does this mean?). If published, this will include your full peer review and any attached files.

Reviewer #1: No

Reviewer #2: No

---

## [Author Response · Author response to Decision Letter 1]

22 Aug 2021

Response to Reviewers

Reviewer #1: 

In introduction section, the authors described “Less than 200 patients with PDE-ALDH7A1 were reported in the literature (5-8).” However, the number of previously reported cases with ALDH7A1 mutations summarized in this paper was biased. In the latest literature, Jiao X, et al. described the number of cases currently reported carrying mutations in the ALDH7A1, which is far greater than described in this paper. (Jiao X, Gong P, Wu Y, Zhang Y, Yang Z. Analysis of the Phenotypic Variability as Well as Impact of Early Diagnosis and Treatment in Six Affected Families With ALDH7A1 Deficiency. Front Genet. 2021;12:644447, and Jiao X, Xue J, Gong P, Wu Y, Zhang Y, Jiang Y, Yang Z. Clinical and genetic features in pyridoxine-dependent epilepsy: a Chinese cohort study. Dev Med Child Neurol. 2020;62(3):315-321.).

Answer: Thanks very much for this question. We added the references and corrected “less than 200” to “about 200”. We appreciate that Jiao et al reviewed the literature and reported in their 2020 manuscript that there were 266 patients reported in the literature. We cited their study following the “About 200 patients with PDE-ALDH7A1 were reported in the literature.” The information is included and highlighted yellow. The two references are also included and highlighted yellow.

We respectfully disagree with the Reviewer 1 regarding the number of patients that we are reporting in our study compared to the studies that Reviewer 1 cited above. Our study is not biased for the patient numbers and cannot be compared with other centers’ patient populations. We are not reporting case series for PDE-ALDH7A1 with their clinical, biochemical and genetic features, rather reporting new findings using muscle biopsy and CSF results. The above references cannot be compared with our study results from these point of view.

The above cited studies were performed at the Peking University, Beijing, China. The population of this city is about 22 million. The population of China is about 1.41 billion. Our study is performed at the University of Toronto. The population of Toronto is about 3 million. The population of Canada is about 37 million. Looking at the population of countries and cities, the above studies regarding patient numbers are likely similar. 

Additionally, we reviewed the references provided by Reviewer 1. We summarized the information below focusing the study reported by Jiao et al., Dev Med Child Neurol, 2020. The authors report 33 patients with pyridoxine-dependent epilepsy (31 patients with ALDH7A1 variants and two patients with PLPBP variants). Authors report biochemical features and genotypes of the patients, summarized below. Unfortunately based on the lack of biochemical abnormalities and/or functional characterization of missense variants, this study did not confirm the diagnosis of PDE-ALDH7A1 in the patients. We summarized below the reasons why we made this conclusion: 

1. Biochemical features: Authors measured α-AASA-P6C in plasma and in urine. In the manuscript they report elevated levels of these metabolites in all 15 patients (patient 1, 3, 4, 5, 6, 9, 10, 12, 14, 15, 16, 17, 18, 19, 20) on page 315, in the paragraph starting with “Fifteen patients underwent …………”. They summarized all biochemical features in Table 1. Their reference ranges are listed below the Table 1. I will only focus on the α-AASA-P6C, as this is the primary biochemical marker for PDE-ALDH7A1 and normal pipecolic acid in urine and plasma was reported previously in PDE-ALDH7A1. 

a. Plasma α-AASA-P6C reference range 4.40–24.17 μmol/L. The 15 patients reported in Table 1 had levels ranging from 4.93-24.17 μmol/L. It seems that based on the reference ranges authors provided below Table 1, all patients had normal plasma α-AASA-P6C levels. 

b. Urine α-AASA-P6C reference range 3.22–122.00 μmol/L in Table 1. The 15 patients reported in Table 1 had levels ranging from 3.22-122. It seems that based on the reference ranges authors provided, all patients had normal urine α-AASA-P6C levels. 

c. All patients with biallelic pathogenic or likely pathogenic variants in ALDH7A1, even on the long-term pyridoxine therapy have elevated urine α-AASA levels. Normal urine and plasma α-AASA-P6C levels exclude the diagnosis of PDE-ALDH7A1. Normal urine and plasma α-AASA-P6C levels are the biochemical functional characterization of variants. It seems that genotypes that the authors report in their study is not supported to be pathogenic or likely pathogenic due to the normal urine and plasma α-AASA-P6C levels in 15 out of 31 patients. 

2. The variants for 31 patients with PDE-ALDH7A1 are listed in supplementary table 2 (S2). The authors did not discuss in detail for the variant classification. 

a. Patient 2: has two intronic variants: c.871+5G>A (IVS9+5G>A) AND c.1008+1G>A (IVS11+1G>A). There is no metabolite measurement for this patient. Usually, the first two bases of introns are considered as pathogenic or likely pathogenic. There are no functional studies if this c.871+5G>A intronic variant will affect the function of ALDH7A1 protein. In the absence of functional study and no measurements of α-AASA-P6C levels, this genotype does not support the diagnosis of PDE-ALDH7A1. 

b. Patient 7: has two missense variants. There are no biochemical investigations for this patient. There are no functional studies of these missense variants if they will affect the function of ALDH7A1 protein. In the absence of functional studies and in the absence of measurement of α-AASA-P6C levels, this genotype does not support the diagnosis of PDE-ALDH7A1.

c. Patient 8: has one missense variant and one splice site variant. There are no functional studies for the missense variant if they will affect the function of ALDH7A1 protein. In the absence of functional studies, and in the absence of measurement of α-AASA-P6C levels, this genotype does not support the diagnosis of PDE-ALDH7A1.

d. Patient 13: has two missense variants. There are no functional studies for the missense variants if they will affect the function of ALDH7A1 protein. In the absence of functional studies, and in the absence of measurement of α-AASA-P6C levels, this genotype does not support the diagnosis of PDE-ALDH7A1.

e. Patient 21: has two missense variants. There are no functional studies for the missense variants if they will affect the function of ALDH7A1 protein. In the absence of functional studies, and in the absence of measurement of α-AASA-P6C levels, this genotype does not support the diagnosis of PDE-ALDH7A1.

f. Patient 22: has one missense variant and one splice site variant. There are no functional studies for the missense variant if this will affect the function of ALDH7A1 protein. In the absence of functional studies, and in the absence of measurement of α-AASA-P6C levels, this genotype does not support the diagnosis of PDE-ALDH7A1.

g. Patient 23: has two missense variants. There are no functional studies for the missense variants if they will affect the function of ALDH7A1 protein. In the absence of functional studies, and in the absence of measurement of α-AASA-P6C levels, this genotype does not support the diagnosis of PDE-ALDH7A1.

h. Patient 24: has one missense variant and one splice site variant. There are no functional studies for the missense variant if this variant will affect the function of ALDH7A1 protein. In the absence of functional studies, and in the absence of measurement of α-AASA-P6C levels, this genotype does not support the diagnosis of PDE-ALDH7A1.

i. Patient 25: has one splice site variant and one deletion. In the absence of biochemical investigations, this genotype does not confirm the diagnosis of PDE-ALDH7A1.

j. Patient 26: has one missense variant (homozygous). There are no functional studies for the missense variant if this missense variant will affect the function of ALDH7A1 protein. In the absence of functional studies, and in the absence of measurement of α-AASA-P6C levels, this genotype does not support the diagnosis of PDE-ALDH7A1.

k. Patient 27: has one missense variant (homozygous). There are no functional studies for the missense variant if this missense variant will affect the function of ALDH7A1 protein. In the absence of functional studies, and in the absence of measurement of α-AASA-P6C levels, this genotype does not support the diagnosis of PDE-ALDH7A1.

l. Patient 28: has one missense variant (homozygous). There are no functional studies for the missense variant if this missense variant will affect the function of ALDH7A1 protein. In the absence of functional studies, and in the absence of measurement of α-AASA-P6C levels, this genotype does not support the diagnosis of PDE-ALDH7A1.

m. Patient 29: has one missense variant and one deletion. There are no functional studies for the missense variant if this missense variant will affect the function of ALDH7A1 protein. In the absence of functional studies, and in the absence of measurement of α-AASA-P6C levels, this genotype does not support the diagnosis of PDE-ALDH7A1.

n. Patient 30: has two missense variants. There are no functional studies for the missense variants if they will affect the function of ALDH7A1 protein. In the absence of functional studies, and in the absence of measurement of α-AASA-P6C levels, this genotype does not support the diagnosis of PDE-ALDH7A1.

o. Patient 31: has two missense variants. There are no functional studies for the missense variants if they will affect the function of ALDH7A1 protein. In the absence of functional studies, and in the absence of measurement of α-AASA-P6C levels, this genotype does not support the diagnosis of PDE-ALDH7A1.

Reviewer #2: 

I felt that the authors have attempted to address concerns which were raised. However, I do think it is important to point out to your readers that AASA is really not as the aldehyde, but rather the intramolecular cyclized form; J Inherit Metab Dis 2019 May;42(3):565-574. doi: 10.1002/jimd.12059. Epub 2019 Mar 11.

Answer: Thanks very much for this comment and question. All these information is included into the introduction and highlighted yellow. We also included reference into the reference list. We hope that our responses would be satisfactory to Reviewer 2.

---

## [Editor Report · Decision Letter 2]

24 Aug 2021

Is impaired energy production a novel insight into the pathogenesis of pyridoxine-dependent epilepsy due to biallelic variants in ALDH7A1?

PONE-D-20-40994R2

Dear Dr. Mercimek-Andrews,

We’re pleased to inform you that your manuscript has been judged scientifically suitable for publication and will be formally accepted for publication once it meets all outstanding technical requirements.

Kind regards,

Israel Silman

Academic Editor

PLOS ONE
---

## [Editor Report · Acceptance letter]

27 Aug 2021

PONE-D-20-40994R2 

Is impaired energy production a novel insight into the pathogenesis of pyridoxine-dependent epilepsy due to biallelic variants in *ALDH7A1*? 

Dear Dr. Mercimek-Andrews:

I'm pleased to inform you that your manuscript has been deemed suitable for publication in PLOS ONE. Congratulations! Your manuscript is now with our production department. 

Kind regards, 

on behalf of

Prof. Israel Silman 

Academic Editor

PLOS ONE